# A Novel Fixed-Time Trajectory Tracking Strategy of Unmanned Surface Vessel Based on the Fractional Sliding Mode Control Method

**Dong Chen, Jundong Zhang * and Zhongkun Li**

College of Marine Engineering, Dalian Maritime University, Dalian 116026, China;
chendong0077@dlmu.edu.cn (D.C.); Zhongkun77@dlmu.edu.cn (Z.L.)
* Correspondence: zhjundong@dlmu.edu.cn; Tel.: +86-0411-8497-8997

**Abstract:** A novel sliding mode control method is proposed to achieve the trajectory tracking of the Unmanned Surface Vessel (USV) and effectively deal with the unmodeled dynamics and external unknown disturbances. First, a fixed-time fractional-order sliding mode control (FTFOSMC) strategy is proposed, combined with the fixed-time control theory and fractional-order control theory based on the sliding mode control method. The FTFOSMC strategy can improve the convergence velocity of the system, and effectively track the desired path, weakening the "chattering" effect in sliding mode control systems. Second, a fixed-time fractional-order sliding mode control strategy combined with the radial basis function neural network (RBF-FTFOSMC) was designed, which can effectively estimate the lumped uncertainties, such as the disturbance of external wind, wave, and current, and the unmodeled dynamics of the USV model. Then, the stability and effectiveness of the designed control strategy are guaranteed by the Lyapunov theory and the corresponding lemmas. Finally, a rigorous simulation experiment is designed to validate the effectiveness and stability of the proposed control strategy. The simulation results show that the control strategy can effectively achieve trajectory tracking of the USV, reduce the "chattering" phenomenon of sliding mode, and effectively estimate the lumped uncertainties.

**Keywords:** USV; fractional-order theory; fixed-time control theory; sliding mode control; RBF neural network; trajectory tracking

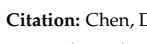



## 1. Introduction

As a representative of the water surface intelligence equipment, the Unmanned Surface Vessel (USV) has a high degree of autonomy and strong advantages in surface target monitoring and reconnaissance [1–4]. As a typical type of uncertainty nonlinear system [5,6], the USV model has the characteristics of nonlinearity, strong coupling, and a large time delay. In addition, it is also susceptible to disturbances such as external wind, waves, and currents during navigation [7]. Therefore, it is of great significance for us to solve the problem of USV trajectory tracking and system disturbances.

In order to solve the USV trajectory tracking problem, many methods, such as PID control [8], model predictive control [9], date-driven control [10,11], and sliding mode variable structure control [12,13], have been continuously developed and applied to trajectory tracking. The sliding mode includes terminal sliding mode control [14], fast integral sliding mode control [15], non-singular terminal sliding mode control [16], and other control methods which are widely used in various intelligent control fields due to their insensitivity to external disturbances and fast response speeds [17]. Qiu [18] proposed a scheme based on the adaptive sliding mode control method for the trajectory tracking of the underactuated USV by using the adaptive and auxiliary dynamic systems to handle unknown disturbances and input saturation, respectively; Lopac [19] proposed a simple sliding mode observation (SMO) to estimate the load angle of a salient-pole wound rotor synchronous

generator, which improves the accuracy during active and reactive power disturbances during stable generator operation; Yang [20] proposed an event-triggered sliding mode control strategy to achieve attitude stabilization on a rigid spacecraft; Liu [21] studied the consensus of the fast sliding mode control method and multi-agent systems, proving that exponential consensus can be achieved in finite-time if the communication network has a directed spanning tree. In the actual control process, the inertia and time delay of the system will continuously cross the sliding mode surface during the reach process, which will lead to the phenomenon of "chattering". All of the above control methods have a slow convergence speed and only guarantee that the system can converge in asymptotic time or finite time. The fixed time control theory was first proposed by Polyakov [22], and this method weakens the "chattering" effect. It also overcomes the shortcomings of the slow convergence speed and ensures that the system convergence time is independent on the initial state. Fractional calculus theory was first proposed by Manabe [23], which is an extension of any order of differentiation and integration. It has the advantages of high freedom of parameter adjustment, rapid response, and low overshoot. Oustaloup and other scholars [24,25] initially applied the theory to intelligence equipment and proved its effectiveness. The fractional-order algorithm is still in its infancy and has not been widely used in the field of USV. In recent years, fractional-order control theory has been relatively widely used in intelligence applications. Jun [26] designed a fractional-order PID (FOPID) control strategy, which effectively improved the motion performance of AUV; Wang [27] proposed a fractional-order motor model of a DC motor and optimized the parameters of a FOPID through the particle swarm algorithm and eventually concluded that after tuning the parameters of FOPID, the fractional-order motor model was better than the integer-order model, and FOPID had a better control effect; Jun [28] combined a recurrent neural network (RNN) and a FOSMC method proposed a RNNFOSMC control strategy, which improves compensation performance and robustness to the active power filter. The fractional-order control theory is still in its infancy and has not been widely used in the field of USV. This paper intends to design a fractional-order sliding mode surface (FOSS) by combining fractional-order control theory and sliding-mode control theory. Then design a fixed-time fractional-order sliding-mode control (FTFOSMC) strategy based on the fixed-time theory. It can improve the convergence speed and reduce the impact of "chattering" on the sliding mode surface, so that the USV can quickly converge to track the desired trajectory and on the initial state of the system.

There will always be external disturbances and unmodeled dynamics that affect the navigation of the USV. In order to effectively solve these problems, there are many control methods that can deal with the system's lumped uncertainties such as the active disturbance rejection control [29], the disturbance observer [30–32], and the neural network control [33]. The neural network control method has been used since the 1940s, and is widely used in various fields due to its precise approximation characteristics and parallel computational capabilities. Wang [34] proposed an adaptive dynamic surface algorithm for collaborative trajectory by combining the dynamic surface control and the single hidden layer neural network. The algorithm realized the estimation of the unknown external disturbances. Zhang [35] used the neural network algorithm to solve the unknown wind, wave, and current disturbance in the system, which effectively realized the path tracking control of the USV. Ignacio [36] designed an adaptive controller for AUV based on deep reinforcement learning and solved the tracking of AUVs control problems by estimating the unmodeled dynamic. Although the above papers have effectively achieved an effective approximation of the unknown disturbance, the convergence time is uniformly asymptotically stable, and therefore it takes a long time to complete the convergence. This paper intends to use the RBF neural network to estimate the external disturbance of the USV and the unmodeled dynamics of the system and guarantee system convergence in fixed-time and improve the convergence accuracy of the system.

To sum up what is discussed above, a fixed-time fractional sliding mode control combined with the RBF neural network (RBF-FTFOSMC) strategy is designed to track an

expected trajectory. First, to solve the problem of slow convergence in the US0V trajectory tracking and control process, a fixed-time fractional sliding mode control (FTFOSMC) strategy is designed to enable USV to track the desired trajectory quickly and stably. Second, an RBF-FTFOSMC strategy is proposed to solve the lumped uncertainties of the system. The lumped uncertainties can be accurately approximated online and the weight can be automatically updated. The feasibility and stability of the proposed control strategy are demonstrated theoretically by the Lyapunov stability theory. Finally, compared with the adaptive fast nonsingular terminal sliding mode (AFNTSMC) [37] and nonsingular fixe-time terminal sliding mode (NFTSMC) [38] simulation results illustrate the effectiveness of the proposed control methods. The simulation results show that the control strategy can quickly realize the trajectory tracking of the USV, improve the convergence speed and effectively weaken the influence of "chattering", and the lumped uncertainties can also be estimated accurately.

In summary, the main contributions of this paper are as follows:

1.  To solve the USV trajectory tracking problem, a novel fixed-time fractional-order sliding mode control (FTFOSMC) strategy is proposed and applied. Compared with the AFNTSMC and NFTSMC control strategies, it reduces the number of control parameters and guarantees accurate trajectory tracking control.
2.  Considering the external disturbance and unmodeled dynamics as the lumped uncertainties, a novel fixed-time fractional-order neural network sliding mode control strategy combined with the RBF-NN (RBF-FTFOSMC) is proposed to ensure that the system can converge in fixed time and effectively realize an accurate estimation of lumped uncertainties.

This paper is organized as follows. Section 2 describes some of the necessary preliminaries and problem descriptions. Section 3 proposes the RBF-FTFOSMC control method and demonstrates the stability of the method. Section 4 presents the simulation results to verify the effectiveness of the proposed method. Section 5 presents the conclusions.

## 2. Problem Description and Preliminaries

### 2.1. Preliminaries

**Lemma 1.** *Consider the following nonlinear systems [39]:*

$$\begin{cases} \dot{x}(t) = l(x(t)) \\ x(0) = x_0, \ l(0) = 0, \ x \in U_0 \subset R^n. \end{cases} \tag{1}$$

*where $x = [x_1, x_2 \dots x_n]^T$, $l(\cdot)$ is a continuous nonlinear function defined on the origin neighborhood $U_0$. If the system has the property of negative homogeneity and can be asymptotically stable, the system (1) is finite-time stable. Suppose that there is a positive definite function $V(x)$:*

$$\dot{V}(x) + \mu V^\kappa \leq 0 \tag{2}$$

*where $\mu > 0$, $0 < \kappa < 1$, then the system (1) will be stable in a finite time.*

**Lemma 2.** *The following property applies to fractional-order theory [40]:*

$$_{t_0}D_t^p \left( {}_{t_0}D_t^q g(w(t)) \right) = {}_{t_0}D_t^{p+q} g(w(t)) \tag{3}$$

*where $p$, $q$ is the fractional order, $D$ is the fractional-order operator, $t_0$ and $t$ represent the time, and $g(w(t))$ represents the function about time.*

**Lemma 3.** *If $V$ satisfies the following conditions [41]:*

$$\dot{V} \leq -\gamma V^m - \chi V^n, V(0) = V_0 \tag{4}$$

where $\gamma$, $\chi$, $m$, and $n$ are positive numbers that satisfy $\gamma > 0$, $\chi > 0$, $m > 1$, and $0 < n < 1$, respectively, then the system $\dot{x} = l(t, x)$ will reach the equilibrium in finite-time and the upper limit time is:

$$T \leq T_{max} = \frac{1}{\gamma(m-1)} + \frac{1}{\chi(1-n)} \tag{5}$$

**Remark 1.** *As shown in Equation (1), the convergence time of finite-time stability is related to the initial state $x_0$. From Equation (5), it can be seen that the convergence time of fixed-time stability is only related to the parameters of the controller and independent on the initial state. Therefore, the upper-bound convergence time of the fixed-time control system is independent on the initial state of the system and overcoming the shortcomings of the finite-time stability.*

**Lemma 4.** *Suppose that there is a continuous radially bounded function $V(x)$ that satisfies the following conditions [42]:*

$$V(x) \leq -k_1 V^p(x) - k_2 V^q(x) + \eta_0 \tag{6}$$

where $p > 1$, $0 < q < 1$, and $\eta_0 > 0$. $k_1$ and $k_2$ are positive numbers satisfying $k_1$, $k_2 > 0$. Then the system $\dot{x} = l(t, x)$ is fixed-time stable and the settling-time is bounded by:

$$T \leq T_{max} = \frac{1}{k_1 \bar{\phi}(p-1)} + \frac{1}{k_2 \bar{\phi}(1-q)} \tag{7}$$

where $\phi$ is a normal number and satisfies $0 < \phi < 1$.

### 2.2. Problem Description

As shown in Figure 1, suppose that the USV is in a two-dimensional plane, establishing the hull coordinate system and inertial coordinate system in the two-dimensional plane, where $O_A X_A Y_A$ is the inertial coordinate, and $O_B X_B Y_B$ is the hull coordinate system. In the coordinate system, the curve represents the trajectory of the USV, $\nu$ is the velocity of the USV, $\psi$ is the yaw angle of the ship, $\mu$ is the surge speed, and $v$ is the sway speed.

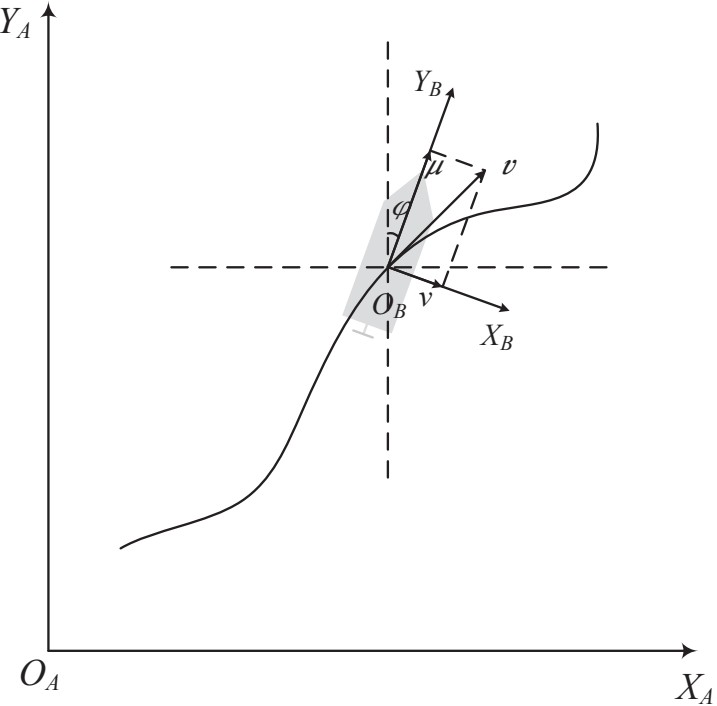

**Figure 1.** USV hull and inertial coordinate system.

The three degrees of freedom kinematics and dynamics equations of the USV in vector form can be expressed as follows [43]:

$$\begin{cases} \dot{\eta} = R(\varphi)v \\ M\dot{v} + C(v)v + Z(v)v = \tau + \delta(t) \end{cases} \tag{8}$$

where the first formula in Equation (8) is the kinematics model of the USV model, where $\eta = [x, y, \varphi]^T$ is the position vector in the inertial coordinate system, $v = [\mu, v, r]^T$ is the speed vector in the hull coordinate system, $R(\varphi)$ is the rotation matrix, used for the mutual rotation between two coordinate systems, and $R(\varphi)$ satisfies the following properties:

$$\begin{aligned} \dot{R} &= RS \\ R^T SR &= RSR^T = S \end{aligned} \tag{9}$$

$$R = \begin{bmatrix} cos(\varphi) & -sin(\varphi) & 0 \\ sin(\varphi) & cos(\varphi) & 0 \\ 0 & 0 & 1 \end{bmatrix} \tag{10}$$

The second formula in Equation (8) is the dynamics model of the USV model, where $\delta(t)$ is the lumped disturbances that include the wind, waves, and currents in the external environment of USV, ans $\tau = [\tau_\mu, \tau_v, \tau_r]^T$ is the control input of USV. Both $C(v)$ and $M(v)$ are constant matrices, where the matrix of Coriolis $C(v)$ is a negative symmetric matrix that satisfies $C(v) = -C^T(v)$. The inertial matrix $M(v)$ is a positive symmetric matrix that satisfies $M = M^T$. They are given as follows:

$$C(v) = \begin{bmatrix} 0 & 0 & c_{13}(v) \\ 0 & 0 & c_{23}(v) \\ -c_{13}(v) & -c_{23}(v) & 0 \end{bmatrix} \tag{11}$$

$$M(v) = \begin{bmatrix} m_{11} & 0 & 0 \\ 0 & m_{22} & m_{23} \\ 0 & m_{32} & m_{23} \end{bmatrix} \tag{12}$$

The nonlinear damping matrix $Z(v)$ satisfies the following:

$$Z(v) = \begin{bmatrix} z_{11}(v) & 0 & 0 \\ 0 & z_{22}(v) & z_{23}(v) \\ 0 & -z_{32}(v) & z_{33}(v) \end{bmatrix} \tag{13}$$

The parameters of the $M$, $C$, and $Z$ are shown in Table 1. Where $m_{ij}$, $i$, $j = 1$, 2, and 3 are the elements of the matrix $M$, and $m$ is the quality of the USV. $X_*$, $Y_*$, and $N_*$ are the derivative of hydrodynamics; $I_z$ is the moment of inertia; $Z_{ij}$, $i$, $j = 1$, 2, and 3 are the elements of the matrix of $Z$ and the elements are related with the hydrodynamic parameters; and $c_{13}(v)$ and $c_{23}(v)$ are the elements of the matrix of $C$, which is related with the velocity and the matrix $M$.

**Table 1.** The parameters of the $M$, $C$, and $Z$.

| Variable Name | Definition | Variable Name | Definition |
|:---:|:---:|:---:|:---:|
| $m_{11}$ | $m - X_{\dot{\mu}}$ | $z_{22}(v)$ | $-Y_v - Y_{|v|v}|v|$ |
| $m_{22}$ | $m - Y_{\dot{v}}$ | $z_{33}(v)$ | $-N_r - N_{|v|r}|v| - N_{|r|r}|r|$ |
| $m_{33}$ | $I_z - N_{\dot{r}}$ | $z_{23}(v)$ | $-Y_r - Y_{|v|r}|v| - Y_{|r|r}|r|$ |
| $m_{23}$ | $mx_g - Y_{\dot{r}}$ | $z_{32}(v)$ | $-N_v - N_{|v|r}|v| - N_{|r|r}|r|$ |
| $m_{32}$ | $mx_g - N_{\dot{r}}$ | $c_{13}(v)$ | $m_{11} - m_{23}r$ |
| $z_{11}(v)$ | $-X_\mu - X_{|\mu|\mu}|\mu|$ | $c_{23}(v)$ | $m_{11}\mu$ |

### 3. Control Strategy Design and Stability Analysis

In this section, based on the theory of fractional-order theory and sliding mode control theory, a fractional-order sliding mode surface (FOSS) is proposed. A fixed-time fractional-order sliding mode control (FTFOSMC) strategy is also proposed combined with the fixed-time control theory and an appropriate sliding mode reaching law. In addition, a novel fixed-time fractional-order sliding mode control strategy combined with the RBF neural network (RBF-FTFOSMC) is proposed which realizes the online approximation of the lumped uncertainties. Finally, the control strategy is guaranteed by the corresponding theory.

*3.1. Model Transformation*

The USV model in Equation (8) can be rewritten with auxiliary variables $x_1 = \eta$, $x_2 = \dot{x}_1$ as follows:

$$\begin{aligned} \dot{x}_1 &= \dot{\eta} = x_2 \\ \dot{x}_2 &= \ddot{\eta} = RM^{-1}\tau + I(\eta, \dot{\eta}) \\ y &= x_1 \end{aligned} \tag{14}$$

where:

$$I(\eta, \dot{\eta}) = RM^{-1}\left[ R^T\delta(t) - (C(\nu) + Z(\nu))R^Tx_2 \right] + Sx_2 \tag{15}$$

The function of (15) is the lumped uncertainty and the lumped uncertainty is continuously differentiable and has an unknown upper bound, which means $||I(\eta, \dot{\eta})|| \leq Z_l$, and $Z_l$ is a positive constant.

*3.2. Fixed-Time Fractional-Order Sliding Mode Control Strategy*

This section proposes the FTFOSMC control strategy, which improves the convergence speed and accuracy of the system, and reduces the "chattering" effect of the sliding mode. First, define the desired position vector $x_r$. The task of the USV is to effectively track the desired trajectory $x_r$. Define $e$ as the USV position tracking error:

$$e = x_1 - x_r \tag{16}$$

According to the position error Equation (16), the fractional-order sliding mode surface (FOSS) $s$ is designed as follows:

$$s = D^{\partial - 1}e + D^{\partial - 2}(\alpha_s\lambda_1(e) + \beta_s\lambda_2(e)) \tag{17}$$

where:

$$\lambda_i(e) = \begin{cases} sig^{\xi_i}(e), & e > \varepsilon \\ \kappa_i e + g_i e^2 sign(e), & e \leq \varepsilon \end{cases} \tag{18}$$

where $\alpha_s$ and $\beta_s$ are positive numbers that satisfy $\alpha_s > 0$ and $\beta_s > 0$, respectively, $\varepsilon$ is a pretty small positive value, $i = 1, 2$; $sig(\cdot)^\delta = |\cdot|^\delta sign(\cdot)$, $\xi_1 = m_1^{sign(|e|-1)}$, $\xi_2 = m_2^{sign(1-|e|)}$, $m_1 > 0, 0 < m_2 < 1$, and the the definition of the $\kappa_i$ and $g_i$ is expressed as follows:

$$\begin{cases} \kappa_i = (2 - \xi_i)\varepsilon^{\xi_i - 1} \\ g_i = (\xi_i - 1)\varepsilon^{\xi_i - 2} \end{cases} \quad i = 1, 2 \tag{19}$$

where $m_1 > 1$ and $0 < m_2 < 1$.

$D^\partial$ is the abbreviation of the fractional-order equation, and the equation satisfies the following:

$$D^\partial f(t) = \frac{1}{\Gamma(z - \partial)} \int_\partial^t \frac{f^{(z)}(\tau)}{(t - \tau)^{\partial - z + 1}} d\tau \tag{20}$$

where $\Gamma(z - \partial)$ is the gamma function with the independent variable $z - \partial$, $z$ is an integer, $D^\partial$ is the fractional-order operator, and $\partial$ satisfies $z-1 < \partial < z$.

The derivative of the FOSS according to Equation (17) satisfies:

$$\dot{s} = D^{\partial-1}\dot{e} + D^{\partial-2}(\alpha_s\dot{\lambda}_1(e) + \beta_s\dot{\lambda}_2(e)) \tag{21}$$

The reaching law of sliding mode surface is designed as:

$$\dot{s} = -\alpha_r sig^{\xi_3}(s) - \beta_r sig^{\xi_4}(s) \tag{22}$$

where $\xi_3 = m_3^{sign(|s|-1)}$, $\xi_4 = m_4^{sign(1-|s|)}$, $m_3 > 1$, $0 < m_4 < 1$, $\alpha_r > 0$, and $\beta_r > 0$. By substituting (21) into (22), the fixed-time fractional-order sliding mode control strategy (FTFOSMC) can be derived as:

$$\tau = MR^{-1}D^{2-\partial}\left[-\alpha_r sig^{\xi_3}(s) - \beta_r sig^{\xi_4}(s)\right] - (\alpha_s\dot{\lambda}_1(e) + \beta_s\dot{\lambda}_2(e)) + \ddot{\eta}_r \tag{23}$$

**Theorem 1.** *Considering that under the designed USV model (14) and error system (16), the proposed control strategy (23) can effectively accomplish trajectory tracking quickly and effectively in fixed-time, with the upper bound of the convergence time is:*

$$T \le \frac{1}{\beta_s(m_1-1)}ln\left(1+\frac{\beta_s}{\alpha_s}\right) + \frac{1}{\alpha_s(1-m_2)}ln\left(1+\frac{\alpha_s}{\beta_s}\right) + \frac{1}{\alpha_r 3^{\frac{1-m_3}{2}} 2^{\frac{1+m_3}{2}}\left(\frac{1+m_3}{2}-1\right)} + \frac{1}{\beta_r 2^{\frac{1+m_4}{2}}\left(1-\frac{1+m_4}{2}\right)} \tag{24}$$

**Proof of Theorem 1.** First, in the stage of the sliding mode arrival, the tracking error $e$ can reach the sliding surface within a fixed time.

Choose the following Lyapunov function:

$$V = \frac{1}{2}s^T s \tag{25}$$

The time derivative of $V$ along with (24) satisfies:

$$
\begin{aligned}
\dot{V} &= s^T\dot{s} \\
&= s^T\left[D^{\partial-1}\dot{e} + D^{\partial-2}(\alpha_s\dot{\lambda}_1(e) + \beta_s\dot{\lambda}_2(e))\right] \\
&= s^T D^{\partial-2}\left[S\dot{\eta} + RM^{-1}\tau - \ddot{\eta}_r\right] + s^T D^{\partial-2}(\alpha_s\dot{\lambda}_1(e) + \beta_s\dot{\lambda}_2(e)) \\
&= s^T\left[-\alpha_r sig^{\xi_3}(s) - \beta_r sig^{\xi_4}(s)\right] \\
&\le -\alpha_r s^T sig^{m_3}(s) - \beta_r s^T sig^{m_4}(s) \\
&\le -\alpha_r 3^{\frac{1-m_3}{2}} 2^{\frac{1+m_3}{2}} V^{\frac{1+m_3}{2}} - \beta_r 2^{\frac{1+m_4}{2}} V^{\frac{1+m_4}{2}}
\end{aligned} \tag{26}
$$

According to Lemma 3, the USV system states can convergence to the FOSS in a fixed time and the upper bound of convergence time is:

$$T_r \le \frac{1}{\alpha_r 3^{\frac{1-m_3}{2}} 2^{\frac{1+m_3}{2}}\left(\frac{1+m_3}{2}-1\right)} + \frac{1}{\beta_r 2^{\frac{1+m_4}{2}}\left(1-\frac{1+m_4}{2}\right)} \tag{27}$$

During the sliding stage, the FOSS satisfies:

$$D^{\partial-1}e + D^{\partial-2}(\alpha_s\lambda_1(e) + \beta_s\lambda_2(e)) = 0 \tag{28}$$

When $|e| < \varepsilon$, we can obtain that:

$$\dot{e} + \alpha_s sig^{\xi_1}(e) + \beta_s sig^{\xi_2}(e) = 0 \tag{29}$$

where $|e|$ satisfies the following:

$$\begin{cases} \dot{e} + \alpha_s sig^{m_1}(e) + \beta_s sig^{\frac{1}{m_2}}(e) = 0, |e| \geq 1 \\ \dot{e} + \alpha_s sig^{\frac{1}{m_1}}(e) + \beta_s sig^{m_2}(e) = 0, |e| < 1 \end{cases} \tag{30}$$

When $|e| \geq 1$, let:

$$U = |e|^{1-m_1} \tag{31}$$

The derivative of $U$ can be written as:

$$\dot{U} = (1-m_1)|e|^{-m_1} esign(e) \tag{32}$$

The Equation (29) can be modified as:

$$\frac{1}{1-m_1}\dot{U} = |e|^{-m_1} esign(e) = -\alpha_r - \beta_r|e|^{\frac{1-m_1 m_2}{m_2}} = -\alpha_r - \beta_r U^{\frac{1-m_1 m_2}{m_2 - m_1 m_2}} \tag{33}$$

According to the above formula, the system on the sliding mode surface will reach the equilibrium point within a fixed time and is independent on the initial state of the system. The upper bound of the convergence time is:

$$T_1 = \frac{1}{m_1-1} \int_0^1 \frac{1}{\alpha_s + \beta_s U^{\frac{1-m_1 m_2}{m_2}}} dU < \frac{1}{\beta_s(m_1-1)} ln\left(1 + \frac{\beta_s}{\alpha_s}\right) \tag{34}$$

It can also be derived that when $|e| < 1$, the system will also reach the equilibrium point in a fixed time and the upper bound of convergence time is:

$$T_2 < \frac{1}{\alpha_s(1-m_2)} ln\left(1 + \frac{\alpha_s}{\beta_s}\right) \tag{35}$$

Therefore, when the system reaches the equilibrium point on the sliding mode surface, the upper bound of the convergence time is:

$$T_s = \frac{1}{\beta_s(m_1-1)} ln\left(1 + \frac{\beta_s}{\alpha_s}\right) + \frac{1}{\alpha_s(1-m_2)} ln\left(1 + \frac{\alpha_s}{\beta_s}\right) \tag{36}$$

Theorem 1 is proven complete. $\square$

**Remark 2.** *The FOSS and FTFOSMC control strategy can be described continuously. During the sliding motion, due to the inertia and the time delay of the system, the sign function will cause the "chattering" phenomenon, while the sat function is continuity, which ensures that the signs will not change suddenly during the sliding stage. Therefore, in order to reduce the impact of the "chattering" better, the sat function will be used instead of the sign function in the stage of reaching of sliding mode control strategy in the actual simulation and experiment process.*

*3.3. RBF-FTFOSMC Control Strategy*

This section will consider the external disturbance of the USV and the unmodeled dynamics of the system. A control strategy combined with the RBF neural network and the FTFOSMC is proposed to estimate the lumped uncertainties online.

The RBF neural network functions can be shown as:

$$\begin{cases} y = W^T h(x) + \varsigma \\ h_i = g\left(\|x - c_i\|^2 / b_i^2\right), i = 1, 2, \ldots, n \end{cases} \tag{37}$$

In function (37), $W^T = [w_1, w_2, w_3, \ldots, w_n]^T$ is the weight of the neural network; $x$ and $y$ denote the input and output vector in the RBF neural network, respectively. The second formula is the Gaussian activation function. $h_i = [h_1, h_2, h_3, \ldots, h_n]^T$ is the linear activation function from the hidden layer to the output layer, $c_i$ is the center of the vector, and $b_i$ is the width of the activation function.

Redefine $f(x)$ as the lumped uncertainties of the ship:

$$f(x) = I(\eta, \dot{\eta}) \tag{38}$$

Define $\hat{f}(x)$ as the estimated value of the RBF neural network function, which is the best approximation function. The estimation error of the lumped uncertainties is

$$\tilde{f}(x) = f^*(x) - \hat{f}(x) \tag{39}$$

Assuming the estimation error is bounded:

$$\tilde{f}_0(x) = sup||f^*(x) - \hat{f}(x)|| \tag{40}$$

Combined with the (39), the control strategy (23) can be rewritten as:

$$\tau = MR^{-1}D^{2-\partial}\left[-\alpha_r sig^{\xi_3}(s) - \beta_r sig^{\xi_4}(s)\right] - (\alpha_s \dot{\lambda}_1(e) + \beta_s \dot{\lambda}_2(e)) + \ddot{\eta}_r - S\dot{\eta} + \hat{f}(x) \tag{41}$$

The time derivative of the FTFOSS satisfies:

$$\begin{aligned}
\dot{s} &= D^{\partial-1}\dot{e} + D^{\partial-2}(\alpha_s \dot{\lambda}_1(e) + \beta_s \dot{\lambda}_2(e)) \\
&= D^{\partial-2}(\ddot{e} + \alpha_s \dot{\lambda}_1(e) + \beta_s \dot{\lambda}_2(e)) \\
&= -\alpha_r sig^{\xi_3}(s) - \beta_r sig^{\xi_4}(s) - D^{\partial-2}\tilde{f}(x)
\end{aligned} \tag{42}$$

where $\tilde{W}$ is the estimated weight of the RBF-FTFOSMC:

$$\tilde{W} = W - \hat{W} \tag{43}$$

The $\dot{\hat{W}}$ can be designed as:

$$\dot{\hat{W}} = \frac{1}{\gamma}D^{\partial-2}sh(x) \tag{44}$$

**Theorem 2.** *When there are lumped uncertainties (15) in the USV model, the RBF-FTFOSMC control strategy (41) can accurately identify the lumped uncertain items within a fixed time, ensuring that the USV can track and converge to an equilibrium position within a fixed time.*

**Proof of Theorem 2.** Choose the following Lyapunov function $L$:

$$L = \frac{1}{2}s^T s + \frac{1}{2}\gamma \tilde{W}^T \tilde{W} \tag{45}$$

The time derivative of $L$ along with (24) and (44) satisfies:

$$
\begin{aligned}
\dot{L} &= s\dot{s} + \gamma \tilde{W}^T \dot{\tilde{W}} \\
&= s^T\left(-\alpha_r sig^{\xi_3}(s) - \beta_r sig^{\xi_4}(s) - D^{\partial-2}\left(\tilde{W}^T h(x) + \varsigma\right)\right) - \gamma \tilde{W}^T \dot{\tilde{W}} \\
&= s^T\left(-\alpha_r sig^{\xi_3}(s) - \beta_r sig^{\xi_4}(s) - \varsigma\right) \\
&\leq -\alpha_r s\, sig^{\xi_3}(s) - \beta_r s\, sig^{\xi_4}(s) \\
&\leq -\alpha_r |s|^{\xi_3+1} - \beta_r |s|^{\xi_4+1} \\
&\leq -\alpha_r \left(|s|^2\right)^{\frac{1+m_3}{2}} - \beta_r \left(|s|^2\right)^{\frac{1+m_4}{2}} \\
&\leq -\alpha_r \left(L - \frac{1}{2}\gamma \tilde{W}^T \tilde{W}\right)^{\frac{m_3+1}{2}} - \beta_r \left(L - \frac{1}{2}\gamma \tilde{W}^T \tilde{W}\right)^{\frac{m_4+1}{2}} \\
&\leq -\alpha_r L^{\frac{m_3+1}{2}} - \beta_r L^{\frac{m_4+1}{2}} + d\left(\tilde{W}\right) + \left(\frac{1}{2}\gamma \tilde{W}^T \tilde{W}\right)^{\frac{m_3+1}{2}}
\end{aligned}
\tag{46}
$$

where $d\left(\tilde{W}\right)$ is the negative expansion and $d\left(\tilde{W}\right) > 0$. According to the Equation (46) and Lemma 4, the system can converge in fixed time and the upper bound of convergence time satisfies:

$$
T_L \leq \frac{1}{\alpha_r \bar{\phi}\left(\frac{m_3+1}{2}-1\right)} + \frac{1}{\beta_r \bar{\phi}\left(1-\frac{m_4+1}{2}\right)}
\tag{47}
$$

The Proof of Theorem 2 is completed. □

## 4. Simulation and Discussion

To validate the effectiveness and stability of the proposed control strategy, numerical simulation will be conducted with the simulation software. The adaptive fast non-singular terminal sliding mode control strategy (AFNTSMC) is selected to compare with the proposed control strategy. Cybership II USV is selected as the simulation object, with the parameters shown in Table 2.

**Table 2.** The parameters of USV.

| Parameters | Values | Parameters | Values | Parameters | Values |
|---|---|---|---|---|---|
| $m$ | 23.8000 | $Y_v$ | −0.8612 | $X_{\dot{\mu}}$ | −2.0 |
| $I_z$ | 1.7600 | $Y_{|v|v}$ | −36.2823 | $Y_{\dot{v}}$ | −10.0 |
| $x_g$ | 0.460 | $Y_r$ | 0.1079 | $Y_{\dot{r}}$ | 0.0 |
| $X_\mu$ | −0.7225 | $N_v$ | 0.1052 | $N_{\dot{v}}$ | 0.0 |
| $X_{|\mu|\mu}$ | −1.3274 | $N_{|v|v}$ | 5.0437 | $N_{\dot{r}}$ | −1.0 |
| $X_{\mu\mu\mu}$ | −5.8664 | | | | |

If the expected initial position vector of the USV is $x_r(0) = [0.1, 0.1, 0]^T$, the actual position is $x_1(0) = [0, 0, 0]^T$, the expected initial speed vector is $v_r(0) = [0, 0, 0]^T$, and the actual ship speed is $v(0) = [0, 0, 0]^T$. In order to ensure the accuracy and validity of the simulation results, the specific control parameters are shown in Table 3; the hidden layer node of the RBF-FTFOSMC control strategy network is 8, the value range of $c_i$ is between $[-1.5, 1.5]$, and the parameter of $b_i$ is set to 3.0.

The disturbance is defined as:

$$
\delta(t) = \begin{bmatrix} \frac{11}{12}(1+0.1\sin(0.2t)) \\ \frac{25}{179}(1+0.3\cos(0.4t)) \\ \frac{950}{636}(1+0.4\sin(0.1t)) \end{bmatrix}
\tag{48}
$$

**Table 3.** The control parameters of AFNTSMC and FTFOSMC.

| Control Strategy | Sliding Mode Parameter | Control Rate Parameter |
|---|---|---|
| AFNTSMC | $\alpha_s = 0.2, \beta_s = 0.2$<br>$m_1 = 5, n_1 = 3, q_1 = 5$<br>$p_1 = 9, \varepsilon = 0.1$ | $\alpha_r = 2.3, \beta_r = 2.3$<br>$m_2 = 5, n_2 = 3$<br>$q_2 = 7, p_2 = 9$ |
| NFTSMC | $\alpha_s = 0.2, \beta_s = 0.2$<br>$m_1 = 5/3, m_2 = 5/9$<br>$h = 0.1, D = 0.2$ | $\alpha_r = 2.3, \beta_r = 2.3$<br>$n_1 = 5/3, n_2 = 7/9$ |
| FTFOSMC | $\alpha_s = 0.2, \beta_s = 0.2$<br>$m_1 = 5/3, m_2 = 5/9$<br>$\varepsilon = 0.1, \partial = 0.3$ | $\alpha_r = 2.3, \beta_r = 2.3$<br>$m_3 = 5/3, m_4 = 7/9$ |

The simulation results are shown in Figures 2–7. The USV trajectory tracking curve and the trajectory tracking in various directions are depicted in Figures 2 and 3. From the above figures, it is observed that each method can achieve the desired trajectory tracking in the presence of lumped uncertainties. However, compared with the AFNTSMC control strategy and the NFTSMC control strategy, the proposed control strategy in this paper can track control of the desired trajectory faster and more effectively. Combined with the derivation of (47), the proposed control strategy can track the trajectory accurately and the upper limit of the convergence time is 0.55 s, while the AFNTSMC control strategy and the NFTSMC control strategy need at least 5 s to track the desired trajectory. It can be concluded from the simulation results that the RBF-FTFOSMC control strategy improves convergence velocity. The trajectory tracking error curve in Figure 4 shows that the AFNTSMC control strategy has a slower convergence speed and lower control accuracy than the proposed control strategy in the presence of lumped uncertainties. The NFTSMC control strategy has a better convergence velocity than the AFNTSMC control strategy, but has a lower convergence accuracy. The proposed control strategy effectively guarantees that the trajectory tracking error can converge to zero in fixed-time accuracy, which effectively improves the USV tracking speed and control accuracy.

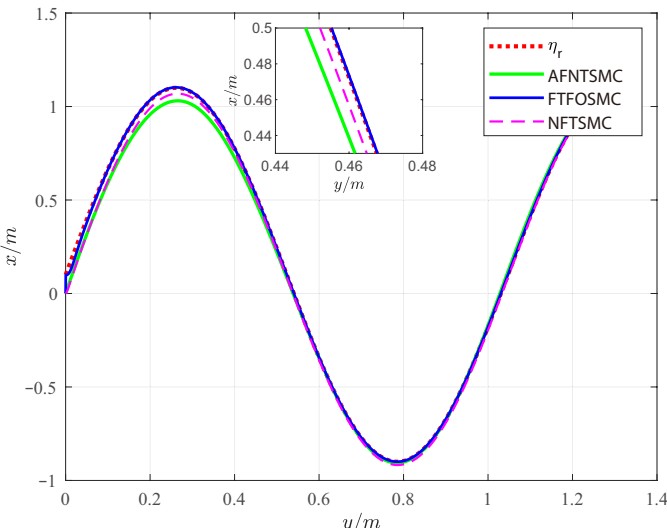

**Figure 2.** USV trajectory tracking curve.

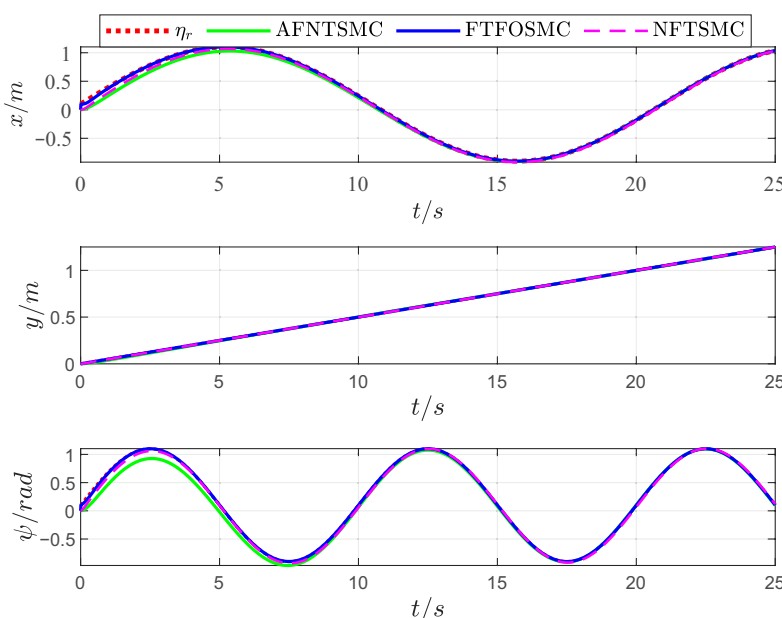

**Figure 3.** USV trajectory tracking curve in all directions.

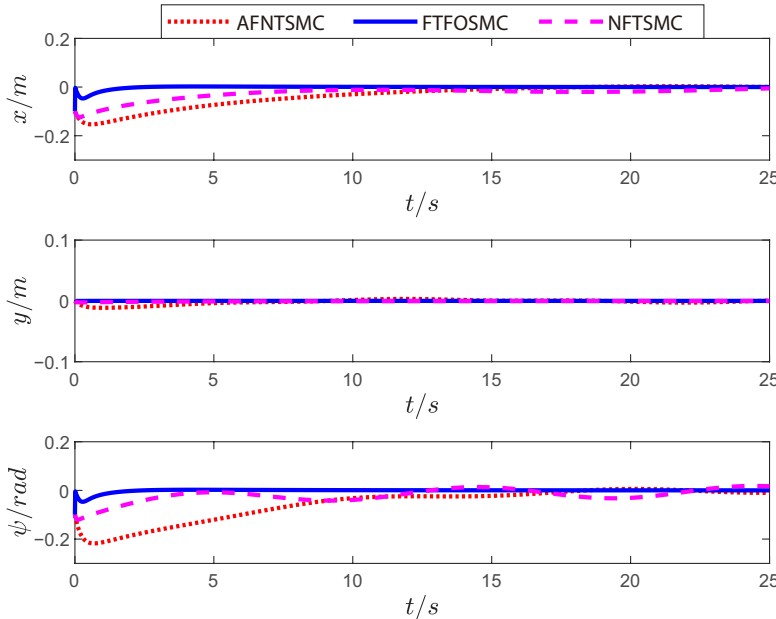

**Figure 4.** USV trajectory tracking error curve.

The speed tracking curve is shown in Figure 5. The three figures show the tracking curves of the surge, sway, and yaw angular speed, respectively. It can be seen that all control strategies can effectively track the velocity of the USV. However, compared with the proposed control strategy in this paper, the NFTSMC control strategy and the AFNTSMC control strategy converge slowly, and the convergence accuracy of the AFNTSMC control strategy is relatively poor. It can be concluded that the proposed control strategy in this paper has faster tracking convergence velocity and accuracy to the desired speed in the presence of uncertainties. Figures 6 and 7 are the control input curves and the estimated curve of the lumped uncertainties, respectively. Figure 6 shows that the control input of the proposed control strategy in this paper has a small overshoot and can achieve the control effect of the trajectory tracking of the USV in fixed-time in the presence of lumped uncertainties. It also reduces the "chattering" phenomenon of the sliding mode control strategy. The robustness of the whole system is effectively improved. In Figure 7, $f_1$ and $f_2$

are the estimation curve of the surge uncertainties and the sway uncertainties, respectively, and $f_3$ is the yaw uncertainties estimation curve of the USV by the RBF neural network. It can be seen that the proposed control strategy in this paper can achieve the estimation of the uncertainties quickly after a short fluctuation in the early stages of the process. It can be concluded that the control strategy proposed in this paper has a faster estimation speed and better estimation effect of the lumped uncertainties, which effectively improves the USV tracking control speed and control accuracy.

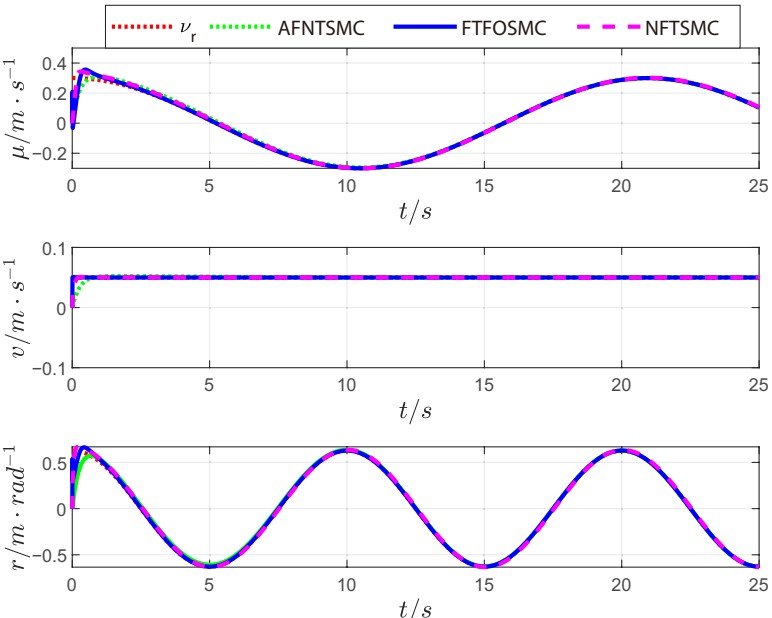

**Figure 5.** Speed tracking curve.

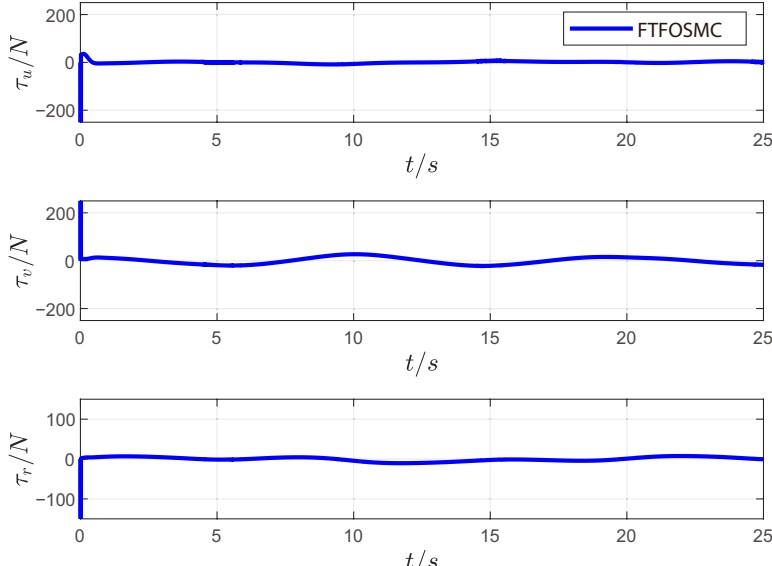

**Figure 6.** Control input curve.

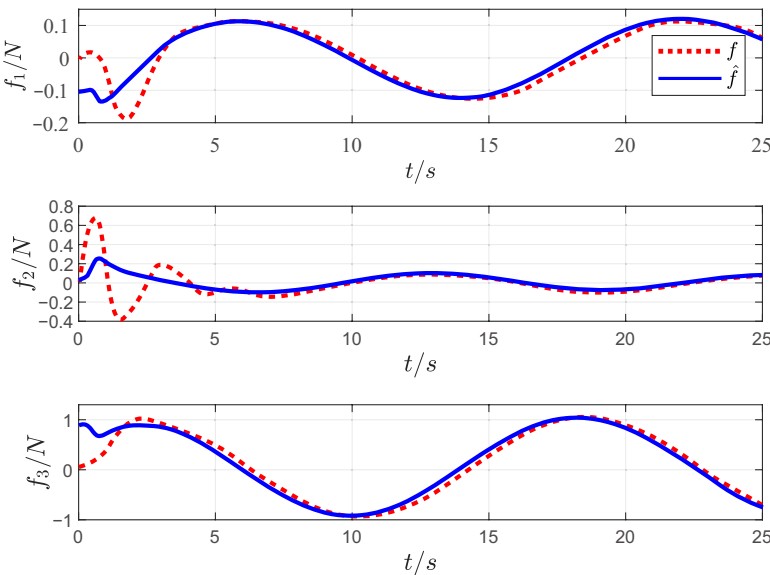

**Figure 7.** Lumped uncertainties estimation curve.

## 5. Conclusions

A novel fixed-time fractional-order sliding mode control strategy was proposed for the trajectory tracking of the USV. This paper proposed sliding mode control surface FOSS that combines fractional-order theory with sliding mode control theory. Additionally, combined with fixed-time control, an appropriate sliding mode reaching law was selected and proposed and a novel control strategy was proposed for trajectory tracking of USV. The FTFOSMC control strategy realized the optimization of the sliding mode control and the accurate tracking of the desired trajectory, effectively improving the convergence speed and reducing the "chattering" phenomenon of the sliding mode. The RBF-FTFOSMC control strategy was proposed to deal with the influence of the lumped uncertainty, which realized the accurate compensation and online approximation of the unmodeled dynamics of the USV model and the unknown external disturbance. Then, the stability was proven according to the corresponding lemma and the Lyapunov theory. Finally, the simulation result shows that the proposed control strategy proposed in this paper can effectively achieve path tracking in fixed-time, and improve the convergence speed and control accuracy.

## 6. Future Recommendation

There are some open questions and limitations that need to be considered. The proposed control strategy is only validated by numerical simulation. Although the USV model has been widely used in many papers, an actual experiment is still needed to provide more rigorous and convincing experimental results. Additionally, the study of the collaborative control between the USV, UAV, and AUV is also worth studying. The fixed-time fractional-order sliding mode control with input saturation can also be considered.

**Author Contributions:** Conceptualization, D.C. and Z.L.; methodology, D.C.; software, D.C.; validation, D.C. and Z.L.; formal analysis, J.Z.; investigation, D.C.; resources, J.Z.; data curation, D.C.; writing—original draft preparation, D.C.; writing—review and editing, D.C.; visualization, Z.L.; supervision, J.Z.; project administration, D.C.; and funding acquisition, J.Z. All authors have read and agreed to the published version of the manuscript.

**Funding:** This research was funded by National Science Foundation of China, grant number U1905212.

**Institutional Review Board Statement:** Not applicable.

**Informed Consent Statement:** Not applicable.

**Data Availability Statement:** The partial data can be found in this paper.

**Conflicts of Interest:** The authors declare no conflict of interest.

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
