# Peer review of "A Novel Fixed-Time Trajectory Tracking Strategy of Unmanned Surface Vessel Based on the Fractional Sliding Mode Control Method"

_electronics, doi:10.3390/electronics11050726_

Round 1

Reviewer 1 Report

The paper is good organized, from abstract to  conclusion. The authors have used the methods for sliding mode control  to achieve the stability of the USV hull and inertial coordinate system.

In  general, this method is very commonly used from other authors, but we suggest the author to use the new method to achieve better  results and compare them with other methods such as for example Neural networks, H∞ etc.

The authors have  used many Lemma, it’s not necessary  to have so many of them.

We have noticed some technical mistakes:

  • Equation 47 to line 144,
  • It is not clear where “ USV model can be rewritten by auxiliary variables x1, x2” which eq. is derived from  ???.

We suggest that  you refer the MDPI Journals “ Bajrami, Xhevahir, et al. "Control Theory Application for Swing Up and Stabilisation of Rotating Inverted Pendulum." Symmetry 13.8 (2021): 1491.”  “El-Fakdi, A.; Cufí, X. An Innovative Low Cost Educational Underwater Robotics Platform for Promoting Engineering Interest among Secondary School Students. Electronics 2022, 11, 373. https://doi.org/10.3390/electronics11030373”

Reviewer 2 Report

This paper proposes a novel fixed-time trajectory tracking strategy based on the fractional-order sliding mode control method for the problem of trajectory tracking of the USV. Besides, the RBF neural network is applied to solve the lumped uncertainty items. The experimental results show that the proposed control strategy is helpful to improve the accuracy and stability of the trajectory tracking of the USV. The overall idea of the article is relatively clear, and the control strategy is relatively novel. The paper is well written and the results are real and effective combined with the comparative experiment. There are some other suggestions for further improvement would be:

  1. The motivation can be enhanced by covering recently published works, such as, DOI: 10.1109/TVT.2021.3136670; IEEE TNNLS, 2021, 32(12): 5456-5467; IEEE/ASME TMech, 2021, 26(6): 3198-3210.
  2. There is something wrong in the equation of “M(v) is a positive symmetric matrix that satisfies M = M(T)”.
  3. Figure 7 could be replaced by a similar format and line colors as above the figure.
  4. The references of 4(209), 17(line 234), 23(line 246) need to be revised.

Reviewer 3 Report

comment:

  1. English writing must be significantly improved. Please scan through the manuscript and carefully check for grammar errors, incomplete sentences, and improper word usage. For example, the “control” in line 4 should be removed, “RBF-FTFOSMC” is not abbreviation of “Fixed-time Fraction-order Sliding Mode Control”, “lemma” in line 11 should be complex, etc..
  2. The theoretical contributions should be stressed in detail in Introduction.
  3. The expression of Lemma 1 is not clear. And in Lemma 2, the meanings of the letters should be given.
  4. The noun description in the text should be unified. For example, “the ship coordinate system” and “the hull coordinate system”.
  5. In Figure 1, the author should mark and explain its meaning.
  6. From equation (12), M matrix is not a matrix about , so should be written as
  7. The author should explain the meaning of the characters in Table 1. For example, , , , etc..
  8. It is obviously unreasonable to include an unknown term in the controller (23).
  9. Theorem 1 and Theorem 2 should be labeled with the formula of the corresponding system or control strategy.
  10. In Remark 2, “In order to reduce the impact of the "chattering" better, the sat function will be used instead of the sign function in the stage of reaching of sliding mode control strategy in the actual simulation and experiment process.” Why don't the author design the controller directly with sat function. Whether using sat function will bring new problems in controller design.
  11. It is suggested that the author should analyze theoretically why The FOSS and FTFOSMC control strategy can reduce “chattering”.
  12. The author should give the type of lumped uncertainties considered in this paper, for example, has unknown upper bound or known upper bound, or the derivative of  has known upper bound or unknown upper bound, etc.
  13. The time series for the example are too long. Using only first 20 seconds would be fairly enough to show the convergence properties of the different control approaches. More than 75% of the time in each graph does not provide any information at all.
  14. Advantages of the proposed algorithm upon the well-known algorithms should be stressed. I suggest comparing the simulations with the results of the recent (2020-2021) related valid references.
  15. The directions to further and improve the work should be added as future recommendation section after ‘conclusions’ section.

Reviewer 4 Report

The manuscript entitled “A Novel Fixed-time Trajectory Tracking Strategy of Unmanned Surface Vessel Based on the Fractional Sliding Mode Control Method” proposes a novel sliding mode control (SMC) method for Unmanned Surface Vessels (USVs). The manuscript proposes the Fixed-time Fractional-order sliding mode control (FTFOSMC) strategy combined with the Fixed-time control and fractional calculus theory based on the SMC method. Moreover, FTFOSMC is also combined with the radial basis function (RBF) neural network. The proposed method is validated by simulations.

The manuscript is well written and easy to follow, with the scientifically sound mathematical framework described in detail. The contributions of the paper are clearly stated. The stability and effectiveness of the proposed control strategy are proved. The results obtained by simulations are clearly presented and appropriately discussed.

However, there are some comments I would like the authors to address before the manuscript is considered for publication:

  1. The research presented in the manuscript is placed within the narrow research field. However, the provided literature review is not very extensive, and some of the references are not recent. Moreover, sliding mode control is a hot research topic with various and diverse applications. Therefore, I would like the authors to extend the literature overview of the sliding mode control research to provide an interested reader with recent applications in various fields. Please consider adding the following papers:
    • doi: 10.1109/TSMC.2018.2867061.
    • doi: 10.1109/TSMC.2018.2816060
    • doi: 10.1109/TCYB.2019.2902868
    • doi: 10.3390/en12091609
    • doi: 10.3390/app9061240
    • doi: 10.1109/TIE.2019.2946541
  2. The proposed control strategy is validated only by simulations. The actual usefulness and applicability of the control method can often only be verified by experiments. Can the authors consider implementing the method in real-time and testing it on a real system to provide experimental verification?
  3. In the Conclusion section, please provide some limitations of the proposed research.
  4. In the Conclusion section, please also provide some challenges and directions for future research.

Round 2

Reviewer 4 Report

The authors have appropriately addressed my comments.
